# Thermoresponsive Polypeptoids

**DOI:** 10.3390/polym12122973

**Published:** 2020-12-12

**Authors:** Dandan Liu, Jing Sun

**Affiliations:** Key Laboratory of Biobased Polymer Materials, Shandong Provincial Education Department, School of Polymer Science and Engineering, Qingdao University of Science and Technology, Qingdao 266042, China; Liudandan0305@126.com

**Keywords:** polypeptoids, thermoresponsive, solid-phase synthesis, ring-opening polymerization (ROP), post-polymerization modification

## Abstract

Stimuli-responsive polymers have been widely studied in many applications such as biomedicine, nanotechnology, and catalysis. Temperature is one of the most commonly used external triggers, which can be highly controlled with excellent reversibility. Thermoresponsive polymers exhibiting a reversible phase transition in a controlled manner to temperature are a promising class of smart polymers that have been widely studied. The phase transition behavior can be tuned by polymer architectures, chain-end, and various functional groups. Particularly, thermoresponsive polypeptoid is a type of promising material that has drawn growing interest because of its excellent biocompatibility, biodegradability, and bioactivity. This paper summarizes the recent advances of thermoresponsive polypeptoids, including the synthetic methods and functional groups as well as their applications.

## 1. Introduction

Stimuli-responsive polymers that can undergo chemical or physical property changes as triggered by a variety of environmental stimuli have been extensively investigated because of their potential applications in the past few decades [1,2,3]. Among the external stimuli, temperature is particularly facile and highly controllable. Thermoresponsive polymers exhibiting a reversible phase transition to temperature are a promising class of smart polymers that have received much attention for various applications such as drug delivery, responsive nanoreactors, and smart hydrogels [4,5,6]. Generally, there are two types of thermoresponsive polymers including lower critical solution temperature (LCST) type and upper critical solution temperature (UCST) type [7]. The LCST type polymer undergoes a phase transition from hydration to dehydration with temperature increasing as a result of the hydrophilic/hydrophobic balance between polymer chains. In contrast, the polymer with UCST properties exhibits opposite phase transition upon heating, where the strong polymer–polymer interactions are formed in aqueous solutions below a critical temperature that can be disrupted with temperature increasing resulted in the dissolution of polymers. In both cases, the phase behavior is dominated by the balance between polymer–polymer and polymer–solution medium interactions that can be finely tuned by temperature. Such temperature-dependent behavior is highly dependent on the inherent properties of the polymers such as the molecular weight, chemical composition, and the solution properties including the concentration and additives in solution [8]. In most cases, the thermoresponsive polymers show typical lower critical solution temperature (LCST) behavior, such as poly(*N*-isopropylacrylamide) (PNIPAM) [5,9,10,11,12], oligo(ethylene glycol)-grafted (meth)acrylates polymers [13], poly(alkyloxazolines) [14,15], and elastin-like polypeptides [16]. It is generally accepted that the most widely studied PNIPAM with LCST behavior is a benchmark thermoresponsive polymer. Generally, a discrepancy in the transition temperature can be observed during the heating and cooling measurement, which is known as the thermal hysteresis. Lutz et al. reported that the random copolymers of 2-(2-methoxyethoxy)ethyl methacrylate and oligo(ethylene glycol) methacrylate (P(MEO_2_MA-*co*-OEGMA)) exhibit a very sharp transition with almost no hysteresis. In comparison, PNIPAM shows a very sharp transition upon heating but a broad hysteresis in the cooling process [17]. Instead, only a few UCST type polymers were reported, such as poly(*N*-acryloylglycinamide) (PNAGA) [18,19] and poly(sulfobetaine) [20]. Seuring and Agarwal et al. first reported the UCST behavior of the PNAGA and found that the UCST behavior of PNAGA can be observed in pure water as well as in physiological milieu under the premise of purely nonionic PNAGA [21]. They also demonstrated that glass transition temperature (*T*_g_) can largely influence the hysteresis of polymers. In particular, some of them exhibit both LCST and UCST behaviors with tunable phase transitions [22]. The development of synthetic approaches in polymer chemistry provides great opportunities for a new generation of thermoresponsive materials with a great level of sophistication. Among the synthetic techniques, atom-transfer radical-polymerization (ATRP), reversible addition-fragmentation, and ring-opening polymerization (ROP) have been extensively used to prepare well-defined thermoresponsive polymers. It has been reported that copolymerization is an excellent strategy allowed to prepare thermoresponsive polymers by tuning the types and compositions of monomers [23]. Meanwhile, post-polymerization modification that can incorporate the pendant functional moieties has also become a versatile synthetic approach to construct polymer with thermoresponsivity [24].

In nature, biomacromolecules such as protein and DNA offer a variety of design principles for stimuli-responsive materials [25,26]. For example, elastin or elastin-mimetic proteins exhibit thermoresponsive properties with tunable LCST for highly biological or catalytic activities of organism living. Considerable efforts have been made to incorporate peptidomimetic polymers into synthetic materials to obtain biohybrid (macro)molecules that mimic proteins in their complexity and functionality. This makes them strong candidates for various applications such as drug delivery, tissue engineering, and gene therapy. Polypeptoids, also known as poly(*N*-substituted glycines), are an emerging class of peptidomimetic polymers with excellent bioactivities for gene transfection, molecular diagnosis, and anti-bacterials [27,28,29]. They have similar backbone composition to polypeptides with *N*-substituted side chains instead of the carbon-substituted pendent groups (Scheme 1). This eliminates the inter- and intra- hydrogen bonds that are intrinsic on polypeptides, and results in more flexible backbones and better solubility in common organic solvents. In addition, the property of the polypeptoids is highly tunable and side-chain dominated, which endows delicate responsive ability to external chemical or physical stimuli.

The polypeptoids are highly designable, where two fundamental synthetic approaches are applied: solid-phase submonomer synthesis and ring-opening polymerization (ROP) of *N*-substituted glycine *N*-carboxyanhydrides (NCA) monomer or *N*-substituted *N*-thiocarboxyanhydrides (NTA) monomer. The peptoid oligomers with short chain length, specific sequence, and monodispersity can be achieved by the solid-phase synthetic method. Such method enables precise control over the functional properties of the peptoids [30,31,32]. In contrast to the solid-phase method, the ROP approach offers an effective way to produce the polypeptoids with high molecular weights and large-scale yields [28,33,34]. In this case, two distinct methods for preparation of functional polypeptoids have been reported including the controlled polymerization of functional monomer and post-polymerization modification (PPM) of well-defined polypeptoid precursors [35]. The polymerization of functionalized monomer method allows for the efficient preparation of the polypeptoids with quantitative functionality and various functional monomers on one individual polymer chain. However, the synthesis of the monomers and controlled polymerization remains a challenge. In contrast, the post-polymerization modification that typically involves “click chemistry” significantly simplifies the synthetic routes, which provides an effective and convenient tool to prepare versatile polypeptoids. In this review, we will mainly discuss recent advances on the design and synthesis of thermoresponsive polypeptoids based on the ROP of NCA or NTA, as well as their biological applications. 

## 2. Synthetic Strategies of Thermoresponsive Polypeptoids

Polypeptoids can be prepared by two fundamental synthetic methods including solid-phase submonomer synthesis and ROP of NCA or NTA. Zuckermann et al. developed a two-step submonomer synthetic method to synthesize peptoids: an acylation step with a haloacetic acid and a displacement reaction with a primary amine, which was an analogy to the well-established Merrifield method of solid-phase peptide synthesis (SPPS) [36,37]. With this approach, high synthesis efficiency and limited chain length, generally up to 50 monomers, have been obtained. In addition, the methods of ROP of NCA or NTA used to synthesize polypeptoids can achieve high molecular weights (MWs) of polypeptoids and large-scale yields [38,39,40]. The synthesis of polypeptoids can be accomplished by the ROP of NCA monomers using primary amine or *N*-heterocyclic carbene (NHC) as initiators. Zhang and coworkers reported NHC-mediated polymerization of NCA monomers, by which polymerization efficiency and MW can be well controlled in solutions with low dielectric constants [41,42]. In addition to *N*-substituted NCAs, ROP of NTA provides a new synthetic method to prepare polypeptoids and in recent years, Ling et al. prepared a series of homopolypeptoids and copolypeptoids with high MWs through controlled ROP of NTA monomers and further explored their physicochemical properties [40,43,44].

### 2.1. The Controlled Polymerization of Functional Monomer Strategy

The polymerization of functional monomer shows prominent advantages to produce the polypeptoids with quantitative modification. In addition, versatile functional monomers can be efficiently incorporated into one individual polymer chain. The polypeptoid monomers for ROP typically include NCA and NTA monomers. A couple of methods have been reported to prepare the monomers [45]. Particularly, NCA monomers can be synthesized from the *N*-substituted glycine precursors in three steps (Scheme 2) [46]. Although the ROP of NCA excludes the activated monomer mechanism, a dimerization reaction occurs to yield 2,5-diketopiperazine particularly at a low concentration of monomers. A type of NTA monomer was reported to polymerize under mild environments tolerant to moisture [47]. The synthetic approach of NTA monomers is similar to that of NCA, except *N*-ethoxythiocarbonyl amino acid is used as the precursor and phosphorous tribromide is used for cyclization reaction instead. Such tunable synthetic approaches greatly promote the development of stimuli-responsive polypeptoid-based materials for many applications. 

### 2.2. Post-Polymerization Modification (PPM) Strategy

Considering the complex purification and protection–deprotection chemistry in the preparation of functional monomer strategy, post-polymerization modification of a well-defined polypeptoid precursor is more convenient. The key of the PPM route is to employ the effective coupling strategy for the pendant moieties to conjugate with the functional groups. “Click chemistry”, such as thiol-ene/yne addition chemistry and copper-catalyzed azide-alkyne click chemistry (CuAAC), has emerged as one of the most efficient strategies to obtain the stimuli-responsive polypeptoids in high yield. Schlaad et al. demonstrated the post modifications of poly(*N*-allyl glycine) (PNAG) and poly(*N*-propargyl glycine) (PNPgG) through thiol-ene chemistry to introduce various functional groups [45,48]. Sun and coworkers successfully prepared a family of thermoresponsive polypeptoids by combining ROP of NCA and thiol-ene/yne click chemistry [49,50,51]. Zhang et al. reported that the cyclic PNPgG homopolymer and copolymer can be modified with azides by CuAAC (Figure 1) [42].

## 3. Thermoresponsive Polypeptoids

Due to the similarity to the chemical structures of a few typical thermoresponsive polymers such as polyacrylamides or poly(2-oxazoline)s, polypeptoids also show lower critical solution temperature (LCST) behavior in aqueous solution. In addition, the chemical structure of the side chains on the polypeptoid polymers can largely determine its thermoresponsive performances. Therefore, precise construction of the side-chain moieties is considered as an efficient strategy to prepare a variety of thermoresponsive polypeptoids. In this section, design and synthesis of thermoresponsive polypeptoids with different side chains will be discussed in detail.

### 3.1. Thermoresponsive Polypeptoids Containing Alkyl Side Chains

Polypeptoids bearing short aliphatic side chains such as methyl (C_1_), ethyl (C_2_), and 2-methoxyethyl *N*-substituents have good solubility while the polypeptoids with longer alkyl side chains (>C_3_) were found to be insoluble in water [52]. Schlaad et al. prepared a series of well-defined poly(*N*-alkyl glycine) with C_3_
*N*-substituents (i.e., *n*-propyl, allyl, propargyl, and isopropyl) by ring-opening polymerization of *N*-substituted glycine *N*-carboxyanhydrides (NCA) and then investigated the solubility of these polypeptoids [53]. They observed that except for the poly(*N*-propargyl glycine), the other three polypeptoids were soluble in water at lower temperature and became insoluble at higher temperature, which showed a typical LCST behavior. It was observed that the phase transition of these polypeptoids can be affected by the concentration, leading to various *T*_cp_. Moreover, the broad transitions and pronounced hysteresis during the heating and cooling scans occurred in the solutions of poly(*N*-propyl glycine) (PNPG) and poly(*N*-isopropyl glycine) (PNiPG), while the PNAG solutions exhibited very sharp transitions that are similar to PNIPAM. In addition, they demonstrated that the molecular weight of polypeptoids show great influence on the cloud point temperatures (*T*_cp_). The *T*_cp_ of PNAG decreases as the molecular weight increases, which may be attributed to a better hydration of shorter chains rather than the contributions of end groups. The opposite trend was observed for PNPG that can be explained by a poor shielding of short hydrophobic backbones in aqueous phase. The *T*_cp_ of the water-soluble polypeptoids was also dependent on the length of the side chains and polymer concentration (Figure 2). They also showed that the crystalline objects can be obtained from the solutions of PNPG and PNAG upon long-time annealing.

Schlaad and coworkers also prepared thermoresponsive poly(*N*-propylglycine)-*block*-poly(*N*-methylglycine) (PNPG-*b*-PNMG) diblock copolypeptoid by successive ring-opening polymerization of NCA-initiated by primary amine [54]. Due to the presence of a thermoresponsive/crystallizable PNPG block and a hydrophilic PNMG block, thermo-induced aggregation occurred in this case. At *T* > *T*_cp_, the initial spherical aggregates morphology transformed into crystallization phase and then larger complex assemblies with flower-like, ellipsoidal, or irregular shapes were obtained (Figure 3).

Zhang and co-workers reported the reversible phase transition of cyclic and linear poly((*N*-ethyl glycine)-*r**an*-(*N*-butyl glycine)) (*c/l*-P(NEG-*r*-NBG)) random copolymers in aqueous solution, both of which were synthesized by ROPs of the respective R-NCA monomers using either NHC or benzyl or butyl amine initiators [55]. It was observed that the cloud point temperature was shifted to higher temperature with increasing of *N*-ethyl glycine content with the *T*_cp_ ranging from 20 to 60℃. The topological architecture (cyclic and linear) has a great impact on the cloud point temperature. The *T*_cp_ of the cyclic copolypeptoids was lower than the linear analogues with the same composition because of the lower entropic loss. It was found that the cloud point temperature decreased with the increase of polymer concentration. The effect of the salt concentrations on the *T*_cp_ of the thermoresponsive polymers was further studied. The results revealed that the *T*_cp_ of copolypeptoids shifted to the lower temperature upon the addition of sodium salts, which was mostly consistent with the Hofmeister series (Figure 4).

Zhang et al. investigated crystallization-driven thermoreversible gelation of coil-crystalline cyclic and linear coil-crystalline diblock copolypeptoids, i.e., poly(*N*-methyl-glycine)-*b**lock*-poly(*N*-decyl-glycine). The cyclic and linear coil-crystalline diblock copolypeptoids were observed to form free-standing gels consisting of entangled fibrils in methanol solutions (5–10 wt%) at room temperature. The gelation dissolved upon heating, which resulted in the transformation of fibrillar network morphology into an isotropic solution that was thermally reversible and mechanically nonreversible (Figure 5) [56]. Further, they synthesized thermoresponsive ABC triblock copolypeptoids (i.e., poly(*N*-allyl glycine)-*b**lock*-poly(*N*-methyl glycine)-*b**lock*-poly(*N*-decylglycine)) that can exhibit sol-to-gel transition at elevated temperature in aqueous solutions at 2.5–10 wt% (Figure 6) [57]. In contrast to the reported ABC triblock copolymers, such as poly(ethylene-*alt*-propylene)-*block*-poly(ethylene oxide)-*block*-poly(*N*-isopropylacrylamide) (PON) and poly((propylenesulfide)-*block*-(*N*,*N*-dimethylacrylamide)-*block*-(*N*-isopropylacrylamide)) (PPS-*b*-PDMA-*b*-PNIPAAM) triblock copolymers, which are based on non-degradable polymers, the prepared ABC triblock copolypeptoids are biodegradable and cytocompatible. The gelation temperature and the mechanical properties of the hydrogels can be finely tuned by regulating the block copolymer composition and the polymer solution concentration.

Thermoresponsive bottlebrush copolypeptoids polynorbonene-*graft*-poly((*N*-ethyl glycine)-*ran*-(*N*-butyl glycine)) (PNor-*g*-P(NEG-*r*-NBG)) comprised of the liner P(NEG-*r*-NBG) random copolymer sidechains were also prepared by the Zhang group. The bottlebrushes were synthesized by the ring-opening metathesis polymerization (ROMP) of the norbornenyl-terminated P(NEG-*r*-NBG) macromonomers [58]. The cloud point transition of the bottlebrush copolymers in aqueous solution is similar to that of the linear macromonomers. However, compared to linear macromonomers, the *T*_cp_ of bottlebrush copolymers was slightly affected by the polymer architecture, but strongly dependent on the thermal history of the solution. On the other hand, the bottlebrush copolypeptoids exhibited the cloud point transition that was irrelevant to the solution thermal history with the addition of inorganic salt. They demonstrated that the cloud point transition was caused by the conformational restructuring of the bottlebrush copolypeptoids facilitated by the thermal annealing (Figure 7). In addition, the presence of salts favored hydrophobic collapse and intermolecular aggregation. Kuroda and Zhang et al. further explored the molecular mechanism of the thermoresponsive performance of polypeptoids by studying two polypeptoids (*l*-NHC-P(NEG-*r*-NBG) and *c*-NHC-P(NEG-*r*-NBG) with similar alkyl side chain compositions and different architecture [59]. The phase transition temperatures of cyclic and linear polypeptoids were 43 °C and 47 °C that were highly dependent on the polymer morphology of macromolecules with very similar solvent interaction but different conformational entropy. This suggested that the phase transition of these polypeptoids is dominated by the conformation of polymer backbone. They further demonstrated the molecular mechanism of the phase transition is associated with the variation of the polymer backbone conformation from a cis amide conformation to a random mixture of cis and trans amide conformations, which is significantly different from the mechanism of the hydration of the polymer resulted by a coil-to-globule transition.

Ling and coworkers synthesized a series of thermoresponsive random copolypeptoids poly(sarcosine-*r**an*-*N*-butylglycine) (P(Sar-*r*-NBG)s) by the controlled copolymerization of sarcosine NTA (Sar-NTA) and *N*-butylglycine NTA (NBG-NTA) initiated by benzylamine [47,60]. They found that P(Sar-*r*-NBG)s show LCST with reversible phase transitions in aqueous solution. The solubility of these copolypeptoids was greatly affected by copolymer composition due to the hydrophobicity of PNBG. The *T*_cp_ can be tuned in a broad temperature range of 27–71 °C by controlling Sar molar fraction and degree of polymerization of random copolymers (Figure 8). It was observed that the *T*_cp_s of polypeptoids displayed a linear increase with the increase of Sar molar fraction, which indicates that the hydrophobic effect plays a crucial role in the thermoresponsive behavior of polypeptoids. Meanwhile, the *T*_cp_ depended slightly on polymer concentration and the MW, as well as salt additives. The *T*_cp_ increased with the decrease of MW, which is consistent with the other vinyl-based polymers [61] and different from the cyclic P(NEG-*r*-NBG)s [55]. Upon the addition of salt additives, the *T*_cp_ decreased, which coincided with the Hofmeister series. Such a result is comparable to the copolypeptoid P(NEG-*r*-NBG)s mentioned previously.

Recently, they synthesized both UV- and thermo-responsive diblock polypeptoids poly(sarcosine-*r**an*-butylglycine)-*b**lock*-PNB (P(Sar-*r*-NBG)-*b*-PNB) and triblock analogs via ROPs of a nitrobenzyl-containing NTA monomer (NB-NTA) with sarcosine NTA (Sar-NTA) and *N*-butylglycine NTA (NBG-NTA) comonomers [62]. The cleavage of nitrobenzyl ester side group occurred after the exposure at 254 nm UV irradiation, which results in the conversion of the lipophilic PNB block to hydrophilic polyiminodiacetic acid block. Simultaneously, the thermoresponsive P(Sar-*r*-NBG) block exhibits LCST phase transition behavior with temperature changing and their *T*_cp_ can be tuned by the composition of repeating units. Additionally, the block polypeptoids can self-assemble in water and form micelles with the PNB blocks as the core because of the amphiphilicity of both the diblock and triblock polypeptoids. Intriguingly, the morphology of spherical nanoparticles transformed into linear aggregates with the increase of concentration and thermal annealing of the samples (Figure 9).

### 3.2. OEGlyated Thermoresponsive Polypeptoids

Poly(ethylene glycol) (PEG) is one of the most commonly used polymers in biotechnology which possesses many excellent properties such as non-toxicity, non-immunogenicity, hydrophilicity, high biocompatibility, and enhanced therapeutic efficacy. Due to the reversible hydration and dehydration of the oligo(ethylene glycol) (OEG) moieties upon temperature change, LCST-type thermoresponsive polymers can be obtained by introducing OEG pendants to the side chain [63]. Aoshima [64] and Lutz [65] investigated a range of biocompatible and thermoresponsive polymers based on (OEG) moieties. The *T*_cp_ of these copolymers could be tuned by adjusting the fraction of OEG units and molecular weight [66,67]. Many pegylated polypeptides have also been extensively investigated, several of them such as poly(L-glutamic acid) and poly(L-cysteine acid) show highly tunable thermal-responsive properties [68,69,70]. However, merely a few of pegylated polypeptides display thermoresponsive properties which may contribute to the fact that the solution properties of OEGylated polypeptides not only depended on substructure of side chains but also relied on their inherent secondary structures. 

A few thermoresponsive polypeptoids have also been successfully prepared by the incorporation of OEG pendants on the side chain. Sun et al. synthesized a series of OEG-grafted poly (*N*-propargyl glycine) (PNPgG_n_-*g*-EG_x_) polypeptoids by a combination of ROP of *N*-propargylglycine *N*-carboxyanhydride initiated by primary amine and thiol-yne click chemistry [49]. They showed that the obtained polypeptoids displayed reversible thermoresponsive behavior in aqueous solution with a sharp phase transition, similar to PNIPAM. This differs distinctly from the previously reported pegylated polypeptoids synthesized by ROP of the corresponding *N*-carboxyanhydrides having oligomeric ethylene glycol side chains, which exhibits good water solubility [71,72,73]. They demonstrated that the introduction of both thioether moieties and OEG on the side chains may vary the amphiphilicity of the system that results in the thermoresponsive property. The influence of degree of polymerization (DP), the side-chain length, the polymer/salt concentration and the chain-ends on the *T*_cp_ was further observed in this work and it was found that the *T*_cp_ of the pegylated polypeptoids could be highly tuned by adjusting these factors (Figure 10). The *T*_cp_ of the polypeptoids increases with the increase of DP, which may be attributed to the short polymer chains that enables increased exposure of hydrophobic backbones to the water phase. In contrast, the polypeptides with similar OEG side-chains exhibited the opposite tendency. This is possibly because with the DP of the polymer increasing, the interaction between polymer chain and water phase reduced due to the increase of the content of a-helix, causing the lower *T*_cp_ [74]. Therefore, it is conceivable that the secondary conformation plays an important role on the thermoresponsive properties of polymers. They utilized different initiators to mediate the polymerization of the monomers and obtained PNPgG_n_-*g*-EG_x_ with different chain ends, polypeptoid with HEXNH_2_ as initator is denoted as HEX-PNPgG_n_-*g*-EG_x_. In all cases, the *T*_cp_s of polypeptoids are comparable, which indicates that the chain-ends have negligible effect on *T*_cp_ [75]. It was also observed that the HEX-PNPgG_n_-*g*-EG_3_ displays a higher *T*_cp_ than that of HEX-PNPgG_n_-*g*-EG_2_ at same DP, which is because of the better hydration ability of the polymer with longer OEG units, irrespective of the backbone structure. Various pegylated polypeptoids can be obtained by adjusting the molar ratios of a mixture of thiol terminated OEG molecules to change the chemical compositions of the pendant side chains. They further synthesized a kind of pegylated polypeptoid that exhibited both reversible thermoresponsive and redox-responsive behaviors in aqueous solution [50]. It was found that the oxidation/reduction of pendant thioethers on the side chain had a great influence on the *T*_cp_ which providing a second stimulus to tune phase transition behavior.

Lately, they reported a stepwise crystallization-driven self-assembly (CDSA) process thermally initiated from amphiphilic poly(*N*-allyl glycine)-*b**lock*-poly(*N*-octyl glycine) (PNAG-*b*-PNOG) diblock copolypeptoid conjugated with thiol-terminated triethylene glycol monomethyl ethers ((PNAG-*g*-EG_3_)-*b*-PNOG) in aqueous solution. The diblock copolypeptoids show a reversible LCST behavior by incorporating of thermosensitive PNAG-*g*-EG_3_ segments, by which the thermostimulus triggers the collapse of hydrophilic chains to promote the orderly fabrication of the crystalline PNOG block upon heating [76]. Due to the presence of a confined domain caused by crystalline PNOG, the morphology transition of the assemblies is irreversible upon a heating-cooling cycle, in which different nanostructured assemblies can be obtained by the same polymer (Figure 11).

### 3.3. Charge-Determined Thermoresponsive Polypeptoids

The thermoresponsive polymers possessing both LCST and UCST show desired dual transitions in a specific range of environmental conditions, which offer great potential for smart materials. However, the polymers that can exhibit tunable phase transitions with both LCST and UCST behaviors in aqueous solution have rarely been reported. This is because delicate control over the balance of a variety of factors is typically required. Wu et al. prepared well-defined thermoresponsive copolymers by copolymerization of *N*-acryloylglycinamide and diacetone acrylamine, which have either LCST or UCSTtype transitions, depending on the compositions, degree of polymerization (DP), polymer concentration, and so on [77]. Wang et al. reported a type of architecture-determined thermoresponsive copolymer with a wide range of controllable LCST and UCST behavior by changing the chain length of poly(vinyl alcohol) grafted on poly(p-dioxanone) [78]. However, such investigations on polypeptoids with dual thermoresponsivity had not been reported until Sun et al. prepared a series of charge-determined ionic polypeptoids with tunable thermoresponsive property by a combination of ring-opening polymerization and click chemistry, which was unlike the previous pegylated nonionic polypeptoids that exhibits LCST behavior [51]. The polypeptoids exhibited either LCST or UCST type behavior, depending on the charge state of the side chains of polypeptoids (Figure 12). Meanwhile, the phase transition of the polypeptoids was also sensitive to PH in aqueous solution which was attributed to the presence of charge. The side-chain architecture, the degree of polymerization (DP), and the polymer concentration were found to have significant effect on the solution properties of the polypeptoids.

In recent work, by introducing alkyl or ethylene glycol (EG) units, a systematic study on thermo-responsive polypeptoid exhibited both LCST and UCST-type behavior was prepared into ionic polypeptoids via combining ROP and thiol-ene/yne click chemistry [79]. The phase transition temperature that is dependent on the chain length of pendant groups, the chemical composition, and the species of ionic moieties can be tuned in the range of 29–55 °C. Intriguingly, they demonstrate that the UCST behavior of polypeptoids can be regulated by the hydrophilic EG group, while the LCST behavior can be tuned by the hydrophobic residues.

## 4. Applications

Various applications of thermoresponsive polymers have been studied in many research fields such as controlled drug delivery, nanoreactors, smart coatings, and sensors [6,80,81,82]. PNIPAM as the most widely studied thermoresponsive polymer has been extensively used in many biomedical applications such as thermoresponsive cell culture dish modified with PNIPAM on which the adhesion and detachment of the cell can be controlled by changing the temperature. The intact cell monolayer can be successfully obtained on the thermoresponsive cell culture dish [83]. Lu et al. synthesized a series of thermoresponsive polypeptides incorporated with various functional side groups [84]. They found that the polypeptides showed excellent biodegradability, biocompatibility, and mechanical properties in in vitro adhesion tests with no cytotoxicity. The polypeptides also showed perfect hemostatic properties and healing effects which are expected to be potential candidates for medical applications, such as tissue adhesives, tissue engineering, and arresting bleeding. Thermoresponsive polypeptoids with temperature-controlled solubility in aqueous solution are promising candidates for biomedical applications due to their good biocompatibility and degradability. Ling et al. investigated the potential of both UV- and thermo-responsive diblock polypeptoids used as a drug delivery system and found that both nile red and pyrene as model hydrophobic guest molecules were encapsulated and released by UV irradiation triggered or a local change in environment [62]. Zhang and coworkers demonstrated that water-soluble enzymes can be easily encapsulated in the ABC triblock copolypeptoid hydrogels for extended period of time with the retained enzymatic activity. The hydrogels show low cytotoxicity towards human adipose-derived stem cells (hASCs) and exhibit bioactivity in modulating the chondrogenesis biomarker gene expression of hASCs, which indicate the potential utilization of the polypeptoid hydrogel as tissue engineering material [57]. The monodisperse, superparamagnetic iron oxide nanoparticles (SPION) modified with a thermoresponsive polypeptoid poly(*N*-methylglyine)-*ran*-poly(*N*-butylglycine) (PNMG-*r*-PNBG) brush was prepared via controlled surface-initiated polymerization of *N*-substituted NCAs [85]. The prepared SPION exhibiting excellent biocompatibility, stability, and high functionality can be used in many applications such as magnetic molecular extraction and magnetic resonance imaging. Garno et al. utilized thiol-ene click reaction to conjugate thermoresponsive copolypeptoid poly((*N*-ethylglycine)_32_-*ran*-(*N*-butylglycine)_17_) P(NEG_32_-*r*-NBG_17_) to 10-undecenyltrichlorosilane (UTS) nanopatterns to prepare nanostructures of copolypeptoid brushes on a silicon substrate [86]. The reversible phase transitions of nanopatterns of copolypeptoids were investigated in situ with atomic force microscopy (AFM) and it was found that the nanostructures shrink in size to form collapsed patterns after heating. On the contrary, the polymer strands stretch out to form taller patterns upon cooling. They demonstrated that the hydrophobic characteristics of surface coatings can be greatly affected by the phase transitions of thermoresponsive polymers anchored to surfaces, which in turn can be used to regulate the adsorption or rejection tendency of the coating to biomacromolecules. Smart coatings of P(NEG_32_-*r*-NBG_17_) with temperature-dependent surface changes will enable the design of surface-based sensors.

## 5. Conclusions

In this review, we summarized the thermoresponsive polypeptoids including the synthetic techniques and functional groups, as well as their applications. The development of synthetic strategies in polymer chemistry results in the facile preparation of thermoresponsive polymers with additional stimulus (e.g., pH, light, enzymes, ionic strength, etc.). This facilitates the synthesis of multi-stimuli responsive polymers with sophisticated behavior and unique properties. A great deal of studies have been devoted to incorporating effective functional groups to polypeptoids to boost their practicability. Compared with other thermoresponsive polymers, thermoresponsive polypeptoids possess excellent biocompatibility, thermal processability, enhanced protease stability, combined with highly tunable phase transition behaviors due to the lack of the backbone chirality and extensive hydrogen bonding interactions in the backbone. There is still much work to do for the further development of thermoresponsive polypeptoids, including developing advanced synthetic strategies for advanced potential applications. In addition, thermoresponsive polypeptoids combined with other stimuli-responsive properties offer great opportunities for the preparation of new generation of smart polymer materials for the development of nanotechnology and material science.

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
