# Peer review of "Thermoresponsive Polypeptoids"

_polymers, 2020, doi:10.3390/polym12122973_

Round 1

Reviewer 1 Report

This is very interesting review, on thermoresponsive polypeptoids. It provides a well-summarized scientific data of this class of materials that exhibit, a reversible response on external stimuli as temperature and/or pH change. The review is generally well written and contains recent advances of thermo-responsive polypeptoids as well as synthetic methods and functional group modifications that strongly influenced their applications. Based on this, I suggest accept after minor revision.

My remarks are:

1) Author should choose which abbreviation to use thermoresponsive as in tittle written or thermo-responsive line 17 as well as stimuili-responsive line 9, 23, 57 etc. 

2.) They should unify the abbreviation of  Poly(N-acryoylglycinamide) (PNAGA) or poly(NAGA).

3.) It would be great benefit for the article acceptance if the in the introduction section  the authors precise the first paragraph when describe the reversibility of the phase transition. It is good to be clarify that some polymers have ultra-rapid volume phase transition at given temperature and at some has hysteresis. 

4.) Figure 2 should  be precise. The symbols a, b, and c are situated on the graph scale. The same with figure 4 and 9. Figure 8 A, B, C should be replaced  with small caps inside instead a b c it should be used numbers. 

5.) Correct, mgmL-1 with mg.mL-1 (Figure 8), Figure 9, Figure 10 ( the authors use mg/mL). Use one designation.

6.) Based on polymer nomenclature the abbreviation -r- on Figure 9 and Figure 1a etc., for the random copolymers is not correct. The correct abbreviation is -ran - and -block- instead -b- (Figure 4a, Figure 6 etc.). I understand, that the authors directly use the corresponding reference abbreviation for the copolymers, but here they are not given and their use  is not correct and informative for the reader. 

7.) Tcp instead Tcp. line 290 etc.

8) Figure 12. The (A) and (B) on the graphics are missing. 

Reviewer 2 Report

This review analyzes studies related to the development of approaches to synthesis, the establishment of properties and practical application of an important class of stimulus-sensitive polymers, thermoresponsive polypeptoids. The authors have provided extensive material on this topic. The article is of undoubted interest for researchers studying thermosensitive polymer materials.

Main remarks.

  1. The review provides a large amount of experimental material, but general patterns are not always discussed in detail.
  2. I propose to slightly correct the reference list. In particular, works [4-6] and [9-12] are relatively old. After their publication, a fairly large number of interesting reviews and breakthrough original articles came out.
  3. In my opinion, in section 3 (Thermoresponsive polypeptoids) discussing the properties of polypeptoids, a comparison with the properties of thermoresponsive polymers of other chemical classes should be made more often in order to highlight the characteristics of polypeptoids.
  4. Lines 158-161. It is necessary to explain why, with an increase in MM, in some cases the cloud point temperature increases, while in others it decreases.
  5. Lines 226-245. Полагаю, что следует более подробно обсудить влияние архитектуры макромолекул на термочувствительность polypeptoids.
  6. Lines 254-257. I propose to describe in more detail the character of the dependence of the cloud point on the MM and the copolymer composition.
  7. In section 3.2 (OEGlyated thermoresponsive polypeptoids) not enough attention is paid to the analysis of the properties of OEGlyated polypeptoids depending on their structure.
  8. Section 4 (Applications) needs to be expanded. In the presented version, only 5 works are discussed in it.
  9. Section 5 (Conclusions) should be slightly corrected, taking into account Notes 1 and 3.
  10. In addition, I would like to see the authors' opinion on the prospects for further research on thermoresponsive polypeptoids.
  11. The manuscript was prepared somewhat carelessly. For example, some abbreviations are entered multiple times (PNIPAM on lines 39 and 42, NNTA on lines 74 and 113, etc.). Poly(N-allyl glycine) uses 2 spellings, namely, poly(N-allyl glycine) and poly(N-allyl glycine) and 2 abbreviations, PNAlG and PNAG. There are other inaccuracies that need to be corrected.
